# Isolation and transcriptomic analysis of *Anopheles gambiae* oenocytes enables the delineation of hydrocarbon biosynthesis

**Linda Grigoraki\*, Xavier Grau-Bové, Henrietta Carrington Yates, Gareth J Lycett, Hilary Ranson\***

Liverpool School of Tropical Medicine, Vector Biology Department, Liverpool, United Kingdom

**Abstract** The surface of insects is coated in cuticular hydrocarbons (CHCs); variations in the composition of this layer affect a range of traits including adaptation to arid environments and defence against pathogens and toxins. In the African malaria vector, *Anopheles gambiae* quantitative and qualitative variance in CHC composition have been associated with speciation, ecological habitat and insecticide resistance. Understanding how these modifications arise will inform us of how mosquitoes are responding to climate change and vector control interventions. CHCs are synthesised in sub-epidermal cells called oenocytes that are very difficult to isolate from surrounding tissues. Here we utilise a transgenic line with fluorescent oenocytes to purify these cells for the first time. Comparative transcriptomics revealed the enrichment of biological processes related to long chain fatty acyl-CoA biosynthesis and elongation of mono-, poly-unsaturated and saturated fatty acids and enabled us to delineate, and partially validate, the hydrocarbon biosynthetic pathway in *An. gambiae*.

**\*For correspondence:**
Linta.Grigoraki@lstmed.ac.uk (LG);
hilary.ranson@lstmed.ac.uk (HR)

**Competing interests:** The authors declare that no competing interests exist.

## Introduction

The cuticle, also known as the exoskeleton, is the outermost part of the insect body and plays a pivotal role in its physiology and ability to adapt and survive in terrestrial environments. The cuticle consists of multiple layers with different composition and properties. The thickest layer, the procuticle, is divided into the endo- and exo-cuticle, both of which are rich in chitin and cuticular proteins. The outer layer, or epi-cuticle, is mainly composed of lipids and hydrocarbons (*Lockey, 1988*). Cuticular hydrocarbons (CHCs) are relatively simple molecules but form complex and varied mixtures of n-alkanes, unsaturated hydrocarbons (alkenes), and terminally and internally methyl-branched alkanes/alkenes. These mixtures of CHCs protect insects from desiccation, are the first barrier to infections from microorganisms and can act as mating recognition signals (pheromones) (*Blomquist, 2010*). The cuticle composition has also been associated with resistance to insecticides, via reduced penetration, in several insect species (reviewed in *Balabanidou et al., 2018*).

*Anopheles* mosquitoes are intensively studied because of their importance as vectors of malaria and lymphatic filariasis that together affect millions of people every year causing intolerable levels of mortality and morbidity. Recently it was shown that populations of the major African malaria vector *Anopheles gambiae* have developed a thicker cuticle with elevated amounts of hydrocarbons and this is associated with a reduction in the penetration rate of pyrethroid insecticides contributing to the high levels of resistance observed (*Balabanidou et al., 2016*). The emergence of pyrethroid resistance is a major concern for vector control strategies as it threatens the efficiency of the insecticide treated nets, all of which contain this insecticide class, that have proven so successful in

**eLife digest** The bodies of insects are encased in an exoskeleton or cuticle that is key for their survival. The cuticle helps protect insects against damage, prevents water loss and can defend against pesticides. A better understanding of the role of the cuticle for survival in mosquitoes and other insects could lead to new ways to prevent the spread of diseases such as malaria.

The cuticle is coated with various molecules from a group of chemicals called hydrocarbons. This coating is made by specialized cells called oenocytes and helps to protect insects. Hydrocarbons can also influence communications between certain insects by acting as recognition signals. In mosquitoes, oenocytes make several hydrocarbons using a set of processes that are not well understood, and the types of hydrocarbons they make can vary between individuals of the same species. It is unclear how this mixture of hydrocarbons is generated and how differences in the mixture can determine how mosquitoes adapt to their surroundings.

Grigoraki et al. studied the genes that were active in isolated oenocytes from the mosquito *Anopheles gambiae*, which carries the parasite that causes malaria. The study revealed a set of genes which are highly active in oenocytes and control the production of fatty acids, a group of molecules used to make hydrocarbons. Other genes involved in creating hydrocarbons were also found. Grigoraki et al. further investigated a specific gene called *FAS1899* and showed that loss of this gene reduces overall hydrocarbon production by 25%. Additionally, genes for transporting and recycling molecules and for producing fats were also shown to be active, which may indicate that oenocytes have a variety of unexplored roles besides making hydrocarbons.

Grigoraki et al. identify the genes involved in producing the hydrocarbon coating of mosquitoes and demonstrate their significance. Further work is needed to understand the precise roles of each of these genes and how they are regulated to adapt the hydrocarbon coating to different situations. This can help explain how the hydrocarbon coating changes in mosquitoes, for example in response to the use of insecticides or climate change. This information is important to adapt and develop new tools to improve mosquito control.

reducing the malaria burden in Africa (*Bhatt et al., 2015*). CHCs are also important in conferring desiccation tolerance in *An. gambiae*, which may be vital in adaptation to arid conditions and survival during the dry season (*Reidenbach et al., 2014*; *Arcaz et al., 2016*).

Cuticular hydrocarbons are synthesised in oenocytes which are secretory cells of ectodermal origin found in most, if not all, pterygote insects (*Makki et al., 2014*). In adult mosquitoes oenocytes are found in characteristic, predominantly ventral, subcuticular clumps that form rows in each segment, while in larval stages they are located in small groups underneath each of the abdominal appendages (*Lycett et al., 2006*).

The biosynthesis of hydrocarbons has been studied using radiolabelled precursors (*Dillwith et al., 1981*) and the biochemical steps of their biosynthetic pathway have been established (*Blomquist, 2010*; *Chung and Carroll, 2015*). The pathway starts with a fatty acid synthase (FAS) that uses malonyl-CoA to generate a fatty acyl-CoA. In the case of methyl-branched hydrocarbons propionyl-CoA groups (as methyl-malonyl-CoA) are also incorporated in the growing fatty acyl-CoA chain. The fatty acyl-CoA chain is further extended by elongases, which extend the chain to different lengths depending on their specificity. Desaturases introduce double bonds, contributing to the generation of unsaturated hydrocarbons, and reductases convert the generated acyl-CoA to aldehydes. These aldehydes serve as substrates for the final step of the pathway, which involves a single carbon chain-shortening conversion to hydrocarbons catalysed by P450 enzymes (*Qiu et al., 2012*). Only this latter step has been delineated in *Anopheles* mosquitoes with two P450 decarbonylases identified, Cyp4G16 and Cyp4G17 (*Balabanidou et al., 2016*; *Kefi et al., 2019*).

Only a subset of the large number of lipid metabolic enzymes encoded in the genome are likely to be significant players in CHC synthesis, but we hypothesised that transcripts from these genes will be specifically enriched in oenocytes to enable this function. Here we report the isolation of oenocytes from adult *An. gambiae* mosquitoes using a transgenic line with fluorescently tagged oenocytes (*Lynd et al., 2019*). RNAseq of the isolated oenocytes identified the key biological processes enriched in these cells and revealed candidate genes for each step of the CHC biosynthetic

pathway. A member of the putative pathway was validated by perturbing expression of the AGAP001899 fatty acid synthase (hereafter called FAS1899). The elucidation of this pathway is a major milestone in delineating the role of variable hydrocarbon composition on key traits that impact vectorial capacity of these important vectors of human disease.

## Results

### FACS isolation of fluorescent oenocytes from transgenic *An. gambiae* mosquitoes

To tag adult *An. gambiae* oenocytes, we expressed the red fluorescent marker m-cherry specifically in these cells using the GAL4/UAS system (*Lynd and Lycett, 2012*). Two transgenic lines were crossed: 1) a homozygous UAS-mCD8: mCherry responder line (*Adolfi et al., 2018*) with 2) a homozygous oenocyte enhancer-GAL4 driver line (Oeno-Gal4) (*Lynd et al., 2019*). Progeny of this cross had the expected m-cherry fluorescent oenocytes throughout their development (*Lynd et al., 2019*). To purify adult oenocytes, mosquitoes were dissected to expose the oenocytes that are dispersed throughout tissues attached to the ventral abdominal integument. Their release was facilitated using trypsin and mechanical homogenisation of the tissue (*Figure 1A*) and subsequent isolation with Fluorescent Activated Cell Sorting (FACS) (*Figure 1B*). Tagged cells corresponded to 1–5% of the total events counted during the FACS sorting and their morphology was consistent with oenocytes by microscopic inspection of sorted cells (*Figure 1—figure supplement 1*).

### Transcriptome analysis of isolated oenocytes and total carcass cells using RNAseq

Triplicate RNAseq libraries were generated using mRNA from isolated tagged cells and total cell populations (cell preparation before FACS, referred herein as carcass cells) from female and male mosquitoes, barcoded and run on the same lane of an Illumina HiSeq sequencer (CGR University of Liverpool). Paired end reads were processed to remove Illumina adapter sequences and low-quality reads. 97.12% of reads passed the quality control and generated a total of 425 million reads, of which 58.1% (+ / - 0.89% standard error) were successfully mapped to the annotated transcripts of *An. gambiae* (Vector Base AgamP4.9).

To visualise how gene expression varied in the different samples we performed a principal component analysis (PCA) using the normalised gene counts of each sample. The first component accounted for 30.1% of the variance in gene expression and separated oenocyte from carcass samples, whereas the second component accounted for 25.9% of variance and reflected differences between females and males. All three replicates of each condition (total female carcass cells, total male carcass cells, female oenocytes, male oenocytes) clustered together (*Figure 1—figure supplement 2*) providing support for robustness of replication between samples.

### Differential expression analysis reveals genes and biological processes enriched in oenocytes

We next identified transcripts significantly over-expressed [$\log_2$(Fold Change)>1, Benjamini-Hochberg adjusted pvalue <0.001, from a Wald test) in oenocytes compared to total (pre-sorted) carcass cells. Our analysis of differential expression identified 1123 genes over-expressed in male oenocytes compared to male carcass cells and 718 genes over-expressed in female oenocytes compared to female carcass cells. From all over-expressed genes 472 were commonly over-expressed in both female and male oenocytes (*Figure 1C* and *Supplementary file 1*). Gene Ontology enrichment analysis for these 472 genes showed an enrichment in biological processes related to sphingolipid biosynthesis, long chain fatty acyl-CoA biosynthesis and elongation of mono-, poly- unsaturated and saturated fatty acids (*Figure 1C*), supporting the role of oenocytes in lipid and hydrocarbon biosynthesis. Other biological processes enriched in the oenocyte samples included endocytic recycling, synaptic vesicle coating and docking, and transmission of nerve impulses. Enrichment analysis of Pfam protein domains showed the over-representation of the ELO family that consists of integral membrane proteins involved in the elongation of fatty acids (*Figure 1C*).

We also investigated whether specific gene isoforms are differentially expressed in oenocytes (at p<0.05, obtained from an empirical cumulative distribution of isoform frequency changes). 672

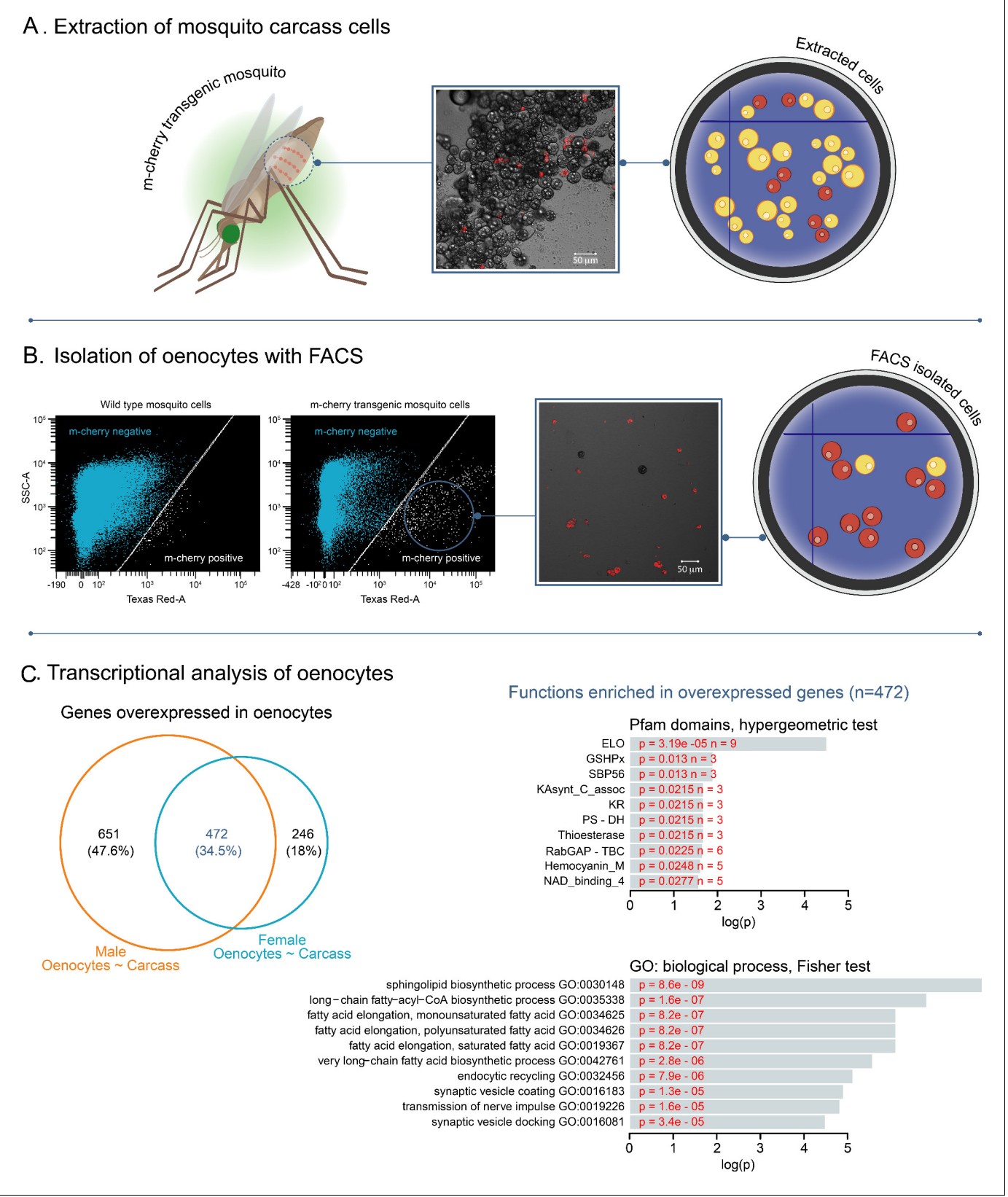

**Figure 1.** Isolation of fluorescently tagged oenocytes and transcriptomic analysis with RNAseq. (**A**) Schematic image of total carcass cells extracted from transgenic *An. gambiae* mosquitoes (progeny of UAS-mCD8: mCherry line and Oeno-Gal4 driver line) expressing the m-cherry fluorescent marker in oenocytes (red cells). (**B**) FACS dot plots. Side-scatter intensity (vertical axis) is plotted against fluorescence intensity (horizontal axis). The sample on

*Figure 1 continued on next page*

*Figure 1 continued*

the left is from wild type G3 mosquitoes, the sample on the right is from transgenic mosquitoes with fluorescent oenocytes. The white line crossing the plots represents the threshold used for sorting mCherry positive cells. (C) Transcriptomic analysis for isolated oenocytes and total carcass cells. Venn diagram for genes over-expressed in both female and male oenocytes vs total carcass cells. Go term (biological process) and Pfam domain enrichment analysis is shown for the 472 genes commonly over-expressed in female and male oenocytes. (ELO: fatty acid elongation, GSHPx: Glutathione Peroxidase, SBP56: Selenium Binding Protein, KAsynt-C: Ketoacyl – synthetase C-terminal extension, KR: KR domain found in polyketide and fatty-acid synthases, PS-DH: Polyketide synthase dehydratase, RabGAP-TBC: RabGTPase-TBC domain).

The online version of this article includes the following figure supplement(s) for figure 1:

**Figure supplement 1.** Representative confocal microscopy image for isolated oenocytes.

**Figure supplement 2.** PCA analysis for the twelve samples used in RNAseq.

genes had at least one isoform differentially expressed in female oenocytes compared to female total carcass cells and 752 have at least one isoform differentially expressed in male oenocytes compare to male total carcass cells. The same analysis was performed for female and male oenocytes showing 578 genes to have at least one isoform differentially expressed between sexes (Appendix 1, *Appendix 1—figure 1*).

## Identification of key candidate genes in the CHC biosynthetic pathway

We next examined which transcripts from members of the six gene families (propionyl-CoA synthases, fatty acid synthetases, elongases, desaturases, reductases and P450 decarbonylases) having roles in the hydrocarbon biosynthetic pathway (*Figure 2*) are differentially expressed in oenocytes. The two P450s, Cyp4G16 (AGAP001076) and Cyp4G17 (AGAP000877), that catalyse the last step in the production of cuticular hydrocarbons, plus the P450 reductase (CPR) that supplies electrons to all P450 monooxygenation reactions, were among the significantly enriched genes (*Supplementary file 1*). Immunolocalisation experiments have previously shown these genes to be highly expressed in *An. gambiae* oenocytes (*Balabanidou et al., 2016*; *Lycett et al., 2006*), lending confidence that our experimental design detects oenocyte enriched genes.

The single propionyl-CoA synthase, AGAP001473, likely responsible for the generation of precursor molecules for the synthesis of methyl-branched hydrocarbons (*Blomquist, 2010*) was enriched in oenocytes. Of the four remaining gene families, specific members were found to be oenocyte enriched; these consisted of three of the four fatty acid synthases (AGAP001899, AGAP08468, AGAP028049), nine of the 20 elongases (AGAP013219, AGAP004372, AGAP001097, AGAP003196, AGAP005512, AGAP007264, AGAP013094, AGAP003195, AGAP003197), one desaturase (AGAP003050 out of nine in the genome) and five of the 17 reductases (AGAP005986, AGAP004787, AGAP005984, AGAP004784, AGAP005985) (*Figure 3*). In addition, the fatty acid transporter AGAP001763, the ortholog of the *Drosophila melanogaster Fatp* (CG7400) functionally implicated in CHC biosynthesis, (*Chiang et al., 2016*) was also enriched in the *An. gambiae* oenocyte transcriptome. The majority of these genes were highly expressed in oenocytes (among the top 200 most highly expressed), with Cyp4G16, Cyp4G17 and FAS1899 (AGAP001899) being in the top ten, followed by the elongase AGAP007264 (*Supplementary files 1*, *2* and *3*).

Interestingly, several of these genes have highly correlated expression. A meta-analysis of 48 transcriptomic datasets from insecticide resistant and susceptible *Anopheles* populations (*Ingham et al., 2018*) identified 44 transcripts co-regulated with Cyp4G16, eight of which were predicted to be part of the CHC pathway. All these eight transcripts, with at least one from each of the six gene families, were present in our experimentally determined CHC synthesising candidate gene list (*Figure 2*).

Notably expression of four genes with a lipid synthesising role was significantly reduced in oenocytes (depicted on *Figure 4*). These include the fatty acid synthase AGAP009176, the desaturase AGAP004572, the reductase AGAP003606 and the elongase AGAP003600. Thus, these genes may be involved in the synthesis of Long Chain Fatty Acids (LCFA) in other tissues, most likely in the fat body, and not specific to the CHC biosynthetic pathway.

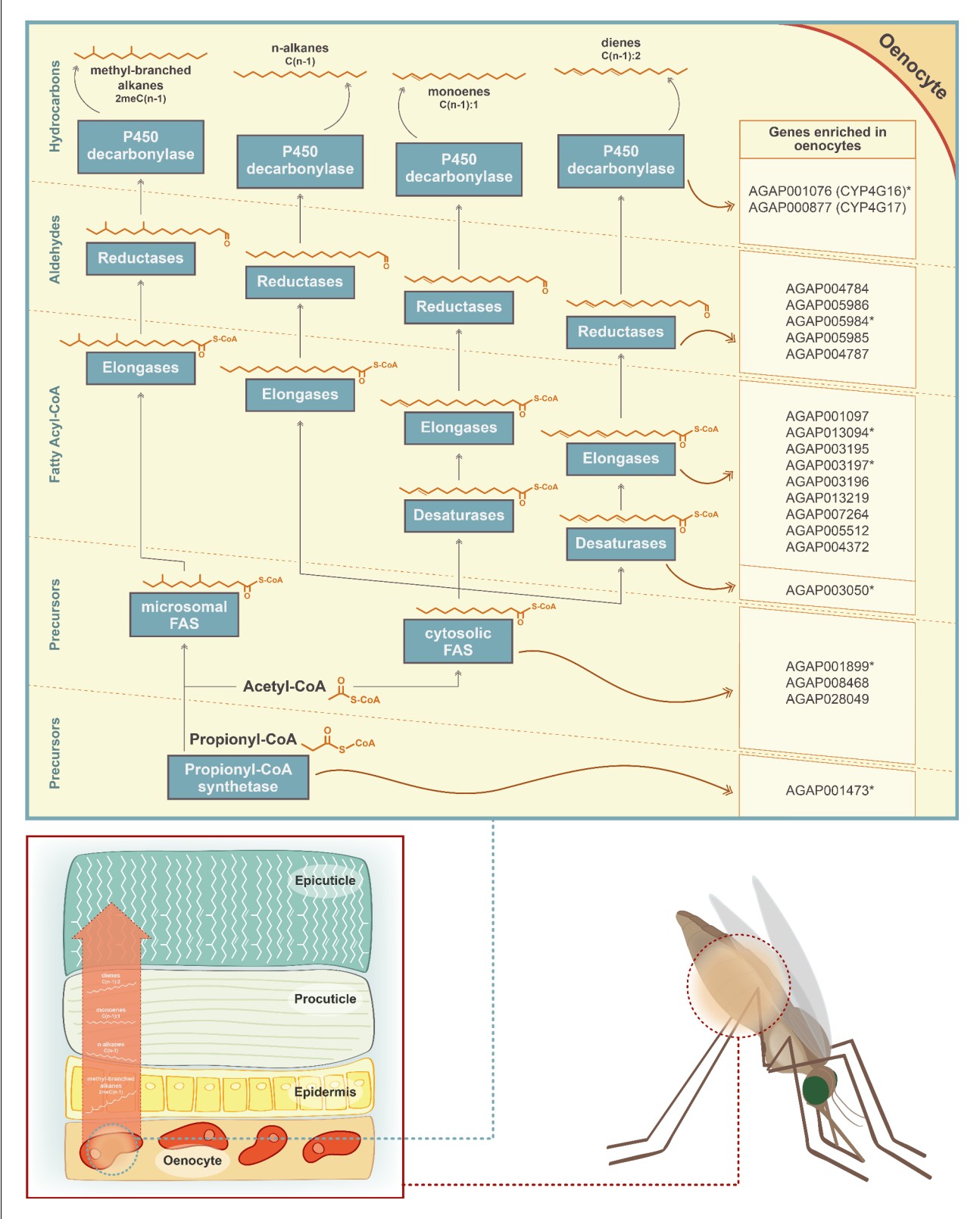

**Figure 2.** Schematic representation of the CHC biosynthetic pathway (adapted from *Chung and Carroll, 2015*). Gene families implicated in the pathway (propionyl-synthetases, fatty acid synthases FAS, elongases, desaturases, fatty acid reductases and decarbonylases) are depicted in blue boxes. The chemical structure of the two precursor molecules of the pathway (Acetyl-CoA and Propionyl-CoA) is shown, as well as the chemical structure of the product of each step of the pathway. Candidate genes for each step of the pathway, with enriched expression in *An. gambiae*

*Figure 2 continued on next page*

oenocytes, are listed on the left. Genes with an asterisk are members of the Cyp4G16 correlation network (*Ingham et al., 2018*). *Note that the epicuticle forms a much thinner layer compared to the procuticle and that its thickness has been enlarged in the image to show the presence of the CHCs.

## Sex specific differential expression analysis of oenocyte expressed genes

We next compared the transcription profile of isolated oenocytes from female versus male mosquitoes. Out of 216 genes that were differentially expressed, 72 were over-expressed in female oenocytes and 144 in male oenocytes. Three genes expressing cuticular proteins (CPR130, CPR25 and CPR26) were significantly and highly (log$_2$FC > 3.9) over-expressed in female oenocytes. However, with the strict criteria we used for the differential expression analysis (log$_2$FC > 1, BH adjusted

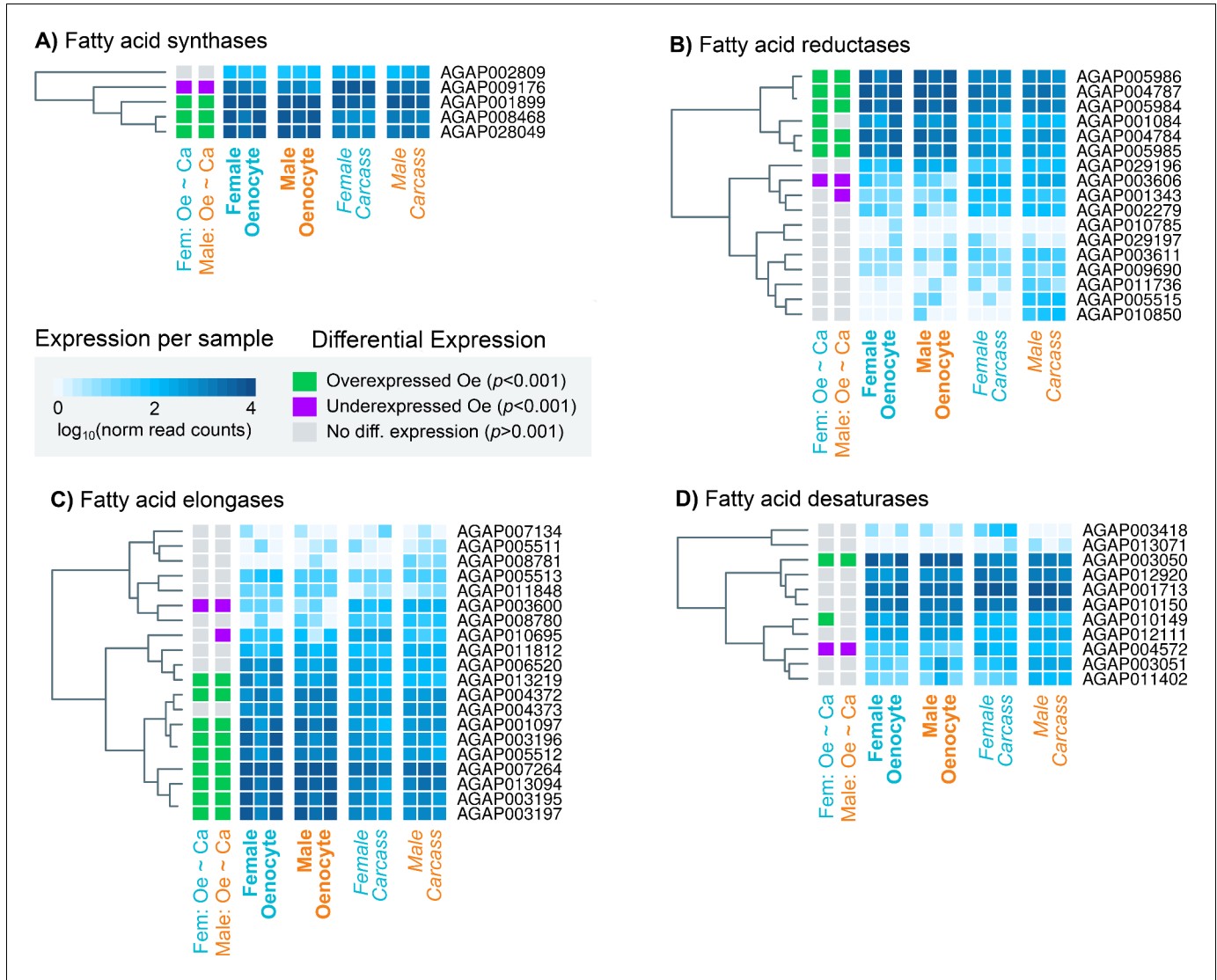

**Figure 3.** Heat maps showing the expression levels of all *An. gambiae* genes belonging to the four gene families (fatty acid synthetases, elongases, desaturases, reductases) implicated in CHC biosynthesis. Expression levels (presented as different intensities of blue and using the log$_{10}$ of the normalised read counts) are shown for all 12 samples used in the RNAseq experiment. The differential expression status in female and male oenocytes vs female and male total carcass cells is shown on the left of each panel. Trees on the left of each map are based on similarities in gene expression. Source data: **Supplementary file 1**.

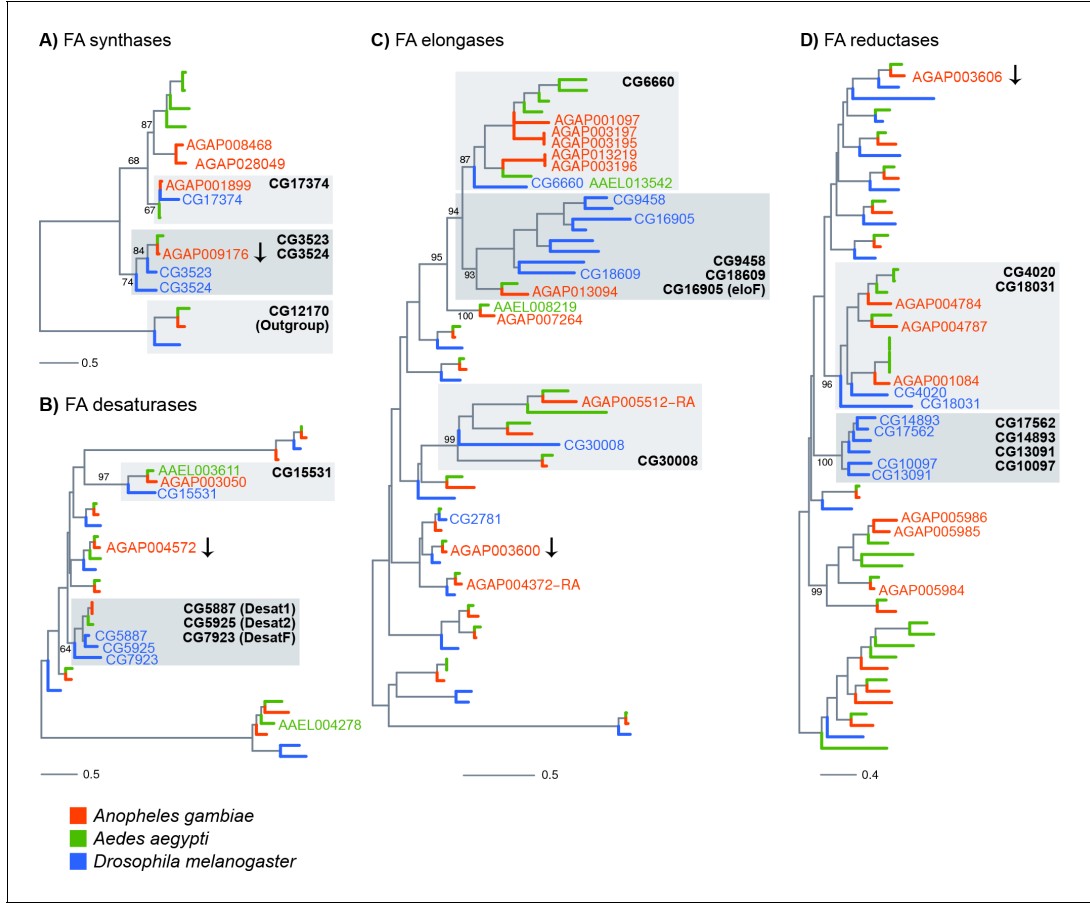

**Figure 4.** Phylogenetic trees constructed for *Anopheles gambiae*, *Aedes aegypti* and *Drosophila melanogaster* genes using the protein domains of Fatty acid synthases (PF00109), desaturases (PF00487), elongases (PF01151) and Fatty acyl-CoA reductases (PF07993). Genes named on trees are: all *An. gambiae* genes found enriched in adult oenocytes (with the exception of genes followed by an arrow, which were significantly down-regulated), *Ae. aegypti* genes found expressed in pupae oenocytes and *D. melanogaster* genes expressed in oenocytes and/or functionally validated (based on provided references). Grey boxes have been added to clades that are discussed in the text and named based on the *D. melanogaster* members. Scale bars show the number of aminoacid substitutions per alignment position. Trees with all gene names are provided in **Figure 4—figure supplements 1–4**.

The online version of this article includes the following figure supplement(s) for figure 4:

**Figure supplement 1.** Phylogenetic tree for *Anopheles gambiae*, *Aedes aegypti* and *Drosophila melanogaster* Fatty acid synthases.

**Figure supplement 2.** Phylogenetic tree for *Anopheles gambiae*, *Aedes aegypti* and *Drosophila melanogaster* Desaturases.

**Figure supplement 3.** Phylogenetic tree for *Anopheles gambiae*, *Aedes aegypti* and *Drosophila melanogaster* Elongases.

**Figure supplement 4.** Phylogenetic tree for *Anopheles gambiae*, *Aedes aegypti* and *Drosophila melanogaster* Fatty acyl-CoA reductases.

p-value<0.001 in all three replicates) we did not find any gene belonging to the hydrocarbon biosynthetic gene families to be differentially expressed between sexes (**Supplementary file 4**).

## Phylogenetic relationships of *An. gambiae* genes implicated in CHC biosynthesis

Phylogenetic trees were constructed for the *An. gambiae*, *Ae. aegypti* and *D. melanogaster* gene families of fatty acid synthases, elongases, desaturases and reductases to provide further insights into gene function, in cases where *Drosophila* orthologs have been characterised, and to identify priority candidates for further study in all three species.

The fatty acid synthases AGAP001899 and AGAP009176 cluster closely with three *Drosophila* FAS genes (**Figure 4A** and **Figure 4—figure supplement 1**), two of which have been shown by in situ hydridisation to be expressed in oenocytes (**Chung et al., 2014**). AGAP001899 is phylogenetically closest to CG17374 (FASN3) known to be expressed in *Drosophila* oenocytes whereas

AGAP009176, the only *An. gambiae* FAS down-regulated in oenocytes, is most closely related to CG3523 (FASN1), which is expressed in the *Drosophila* fat body.

AGAP003050 is the only desaturase enriched in both female and male oenocytes and is a clear 1:1 ortholog of *D. melanogaster* CG15531 (*Figure 4B* and *Figure 4—figure supplement 2*) with a predicted stearoyl-CoA 9-desaturase activity, and AAEL003611 (also found expressed in *Ae. aegypti* oenocytes [*Martins et al., 2011*]). AGAP001713 and AGAP012920, the paralog of the three *Drosophila* desaturases Desat1 (CG5887), Desat2 (CG5925) and Fad 2 (CG7923) involved in the production of unsaturated hydrocarbons (*Dallerac et al., 2000*; *Chertemps et al., 2006*), some of which act as pheromones, were not among the oenocyte enriched genes.

The elongase family appears to have radiated further after evolutionary separation of *Drosophila* and mosquitoes. Five out of the nine *An. gambiae* elongases that are enriched in oenocytes (*Figure 4C* and *Figure 4—figure supplement 3*), (AGAP001097, AGAP003195, AGAP003196, AGAP003197, AGAP013219) form a cluster of paralogs phylogenetically related to a single *Drosophila* elongase, CG6660, a gene over-expressed in adult oenocytes (*Huang et al., 2019*). Two of these paralogs (AGAP003196 and AGAP013219) are closely related to the *Ae. aegypti* AAEL013542 elongase, which is also expressed in pupae oenocytes (*Martins et al., 2011*). AGAP013094, another oenocyte enriched elongase is the single *An. gambiae* gene in a cluster of *D. melanogaster* paralogs with known functions in CHC biosynthesis, such as eloF (CG16905) (*Chertemps et al., 2007*), CG30008, CG18609 and CG9458 (*Dembeck et al., 2015*).

Contrary to the other gene families, most fatty acid reductases enriched in oenocytes did not have clear orthology relationships with functionally characterised *D. melanogaster* genes (*Figure 4D* and *Figure 4—figure supplement 4*). For example, AGA005984, AGA005985 and AGA005986 clustered in a culicine-specific group of paralogs with no *Drosophila* orthologs. Similarly, most of the fly genes that are functionally linked to CHC profiles (*Dembeck et al., 2015*), such as CG13091, CG10097, CG17562 and CG18031, form a cluster of paralogs with no one-to-one orthologs in *An. gambiae*.

## Functional validation of candidate genes

The fatty acid synthase, FAS1899 and the desaturase, Desat3050, both of which were significantly enriched in oenocytes, were selected for functional validation. We knocked-down their expression through oenocyte specific RNAi and examined the effect on the CHC profile.

Firstly UAS-regulated responder lines carrying *FAS1899* and *Desat3050* hairpin RNAi constructs were established. Crossing the responder lines with the oenocyte specific-Gal4 promoter line (Oeno-Gal4) (*Lynd et al., 2019*) resulted in ~80% knock down for the FAS1899 and ~26% for the Desat3050 (in L2 larvae). In both cases oenocyte specific RNAi suppression was lethal at the L2/L3 larvae stages. Subsequently we crossed the two responder lines with the Ubi-A10 Gal4 line (marked by CFP) (*Adolfi et al., 2018*) which directs widespread tissue expression, but at lower levels in oenocytes compared to the oeno-Gal4 line. The majority of progeny from these crosses expressing dsRNA for FAS1899 and Desat3050 reached the pupae stage, but 70–80% died either as mid to late pupae or during adult emergence (*Figure 5—figure supplement 1*). QPCR analysis in whole adults indicated a ~ 26% knock down of FAS1899 transcripts, but no significant difference in Desat3050 knockdown.

GC-MS analysis of the hexane extracted hydrocarbons revealed the presence of at least 60 CHC peaks in all samples; 15 of which were alkanes, five unsaturated alkanes and 40 methyl-branched alkanes. While 19 of the CHC peaks had an abundance of ≥1%, the alkanes C29, C27 and C31, and the methyl-branched methyl-C31 were consistently among the most abundant accounting for approximately half of the total CHCs (*Figure 5—source data 1*).

The CHC profile of surviving FAS1899 and Desat3050 knock down adults was compared to control siblings. A significant (Student's t-test p-value≤0.05) 25% reduction in the total amount of hydrocarbons was observed in both female and male FAS1899i mosquitoes. The proportion of the different CHC categories also changed significantly (Student's t-test p-value≤0.05) in the FAS1899i individuals, as the relative abundance of methyl-branched hydrocarbons decreased while the relative abundance of unsaturated and n-alkanes increased (*Figure 5*). No difference in the total amount of CHCs, nor of the % of unsaturated CHCs, was observed for the surviving Desat3050 knock down adults (*Figure 5—source data 1*).

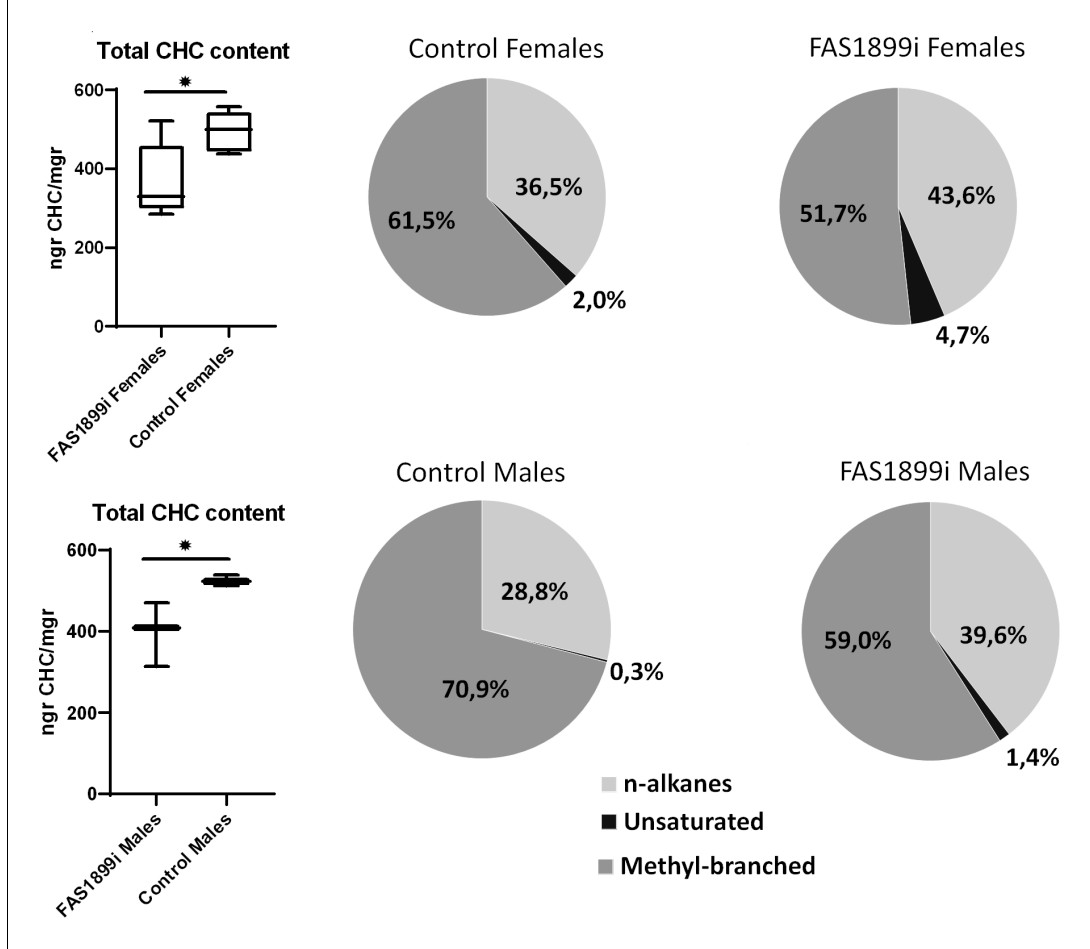

**Figure 5.** Comparison of the total CHC content quantified with GC-MS in female and male adults with knock down of FAS1899 (Ubi-A10 Gal4/UAS-FAS1899i) and control siblings (heterozygous Ubi-A10 Gal4/+). The mean values of total CHC/mgr (± SEM) are: for FAS1899i females 368ngr/mgr, for control females 494ngr/mgr, for FAS1899i males 398ngr/mgr and for control males 525ngr/mgr (five biological replicates for females and three for males). The box plots show the 25th and 75th percentile; the mean is shown as a black line within the box; error bars correspond to the minimum and maximum values. Pie charts represent the relative abundance of the three CHC categories (n-alkanes, unsaturated alkanes and methyl-branched alkanes) in FAS1899i and control individuals. Statistical test performed: Student's t-test (p-value≤0.05), Source data and p-values provided in *Figure 5—source data 1*.

The online version of this article includes the following source data and figure supplement(s) for figure 5:

**Source data 1.** GC-MS analysis of CHCs.

**Figure supplement 1.** Lethality phenotype of progeny from crosses.

## Discussion

CHCs affect key traits in *Anopheles* mosquitoes that determine their fitness and thus vectorial capacity. The difficulties in isolating the CHC synthesising cells in adult mosquitoes, due to their close association with fat body cells within the abdomen, and the absence of clear one to one orthologs with *Drosophila* in some families (*Figure 4*), has hindered the identification of genes involved in mosquito CHC production. In this study we describe the FACS purification of fluorescently tagged oenocytes from adult *An. gambiae* mosquitoes, and the subsequent transcriptomic analysis of the purified cells which enabled us to identify key candidate genes in the CHC biosynthetic pathway.

The samples analysed consisted of total cells recovered from dissected abdomen integument, containing ~12% of tagged oenocyte cells, which were then compared to purified oenocyte cells isolated by passage through the FACS. The abdomen tissue is mainly composed of fat body and epithelial cells, neither of which are expected to synthesise hydrocarbons. Fat bodies do however have a primary role in lipid biosynthesis, which has several steps in common with the CHC biosynthetic

pathway, both utilising fatty acid synthases, elongases and desaturases. The analysis pathway was purposively designed to reveal genes and isoforms that are predominantly enriched in oenocytes and thus likely to be involved in CHC biosynthesis but a limitation, in our goal to delineate the entire CHC pathway, is that it will likely fail to detect genes that are expressed at similar levels in fat bodies and oenocytes and are involved in both CHC and lipid biosynthesis (*Wicker-Thomas et al., 2015*).

Our data set is the first transcriptomic data for adult mosquito oenocytes. Limited depth transcriptional analysis of larval *Ae. aegypti* oenocytes that persist during early pupal development, and are relatively easily dissected in pure form due to their distinct large size and loose attachment as clumps of cells to the integument (*Makki et al., 2014*), has previously been performed (*Martins et al., 2011*). Comparison of the partial oenocyte *Aedes* transcriptome with our adult *Anopheles* oenocyte data set provides insights into key genes potentially involved in CHC synthesis throughout development. Seven genes involved in lipid biosynthesis were detected in *Aedes* larval oenocytes, including one acetyl-coA synthetase (AAEL007283), two elongases (AAEL008219 and AAEL013542), two desaturases (AAEL003611 and AAE004278) and the two orthologs of Cyp4G16 and Cyp4G17 (AAEL004054 and AAEL006824). Clear orthologs for five of these larval oenocyte expressed genes (except for the desaturase AAEL004278) were present in our *An. gambiae* adult oenocyte transcriptome (*Figure 4*). Further work to characterise *Anopheles* oenocyte transcriptomes at earlier life stages will be facilitated by this FACS approach to enable functional analysis of these cells during mosquito development.

In addition to genes involved in lipid and hydrocarbon biosynthesis, genes associated with the biological processes of synaptic vesicle coating and docking, and nerve impulse transmission were found enriched in the oenocyte transcriptome. The Oeno-Gal4 driver line used to generate the mosquito population with fluorescent oenocytes has a red fluorescent marker (dsRed) under the control of the 3xP3 promoter that drives expression in the eyes and nerve cord. A small contamination of the FACS isolated oenocytes with cells of the nerve cord could be speculated, although nerve cells were not observed when visually observing the isolated cells with confocal microscopy. Moreover, oenocytes have been reported to play a role in the neuronal processes during *D. melanogaster* embryogenesis through the secretion of semaphorin (Sema2a), a peptide that drives axon elongation; ablation of oenocytes results in sensory axon defects similar to the *sema2a* mutant phenotype (*Bates and Whitington, 2007*). In addition, the development of oenocytes and of sensory organ precursors (SOPs) in the peripheral nervous system is intimately linked. A previous study has shown that in *D. melanogaster* embryos primary SOPs signal, via the EGFR pathway, to the overlying ectoderm. This results in the differentiation of signal receiving cells into oenocytes, in the presence of the Sal transcription factor, or into secondary SOPs in the absence of Sal (*Rusten et al., 2001*).

Thus, oenocytes likely have a variety of currently unexplored functions, which is also supported by our gene enrichment analysis. A further interesting aspect is their potential role in lipid synthesis, processing and secretion (signalling), as reviewed in *Makki et al., 2014* and suggested by the enrichment in our dataset of sphingolipid and fatty acid biosynthesis and endocytic recycling. Further studies are needed to explore these functions in more detail and to investigate the potential cross talk (regulation) of oenocytes with other tissues. For example *D. melanogaster* larval oenocytes are thought to produce a VLCFA dependent signal that controls remotely the water-tightness of the respiratory system (*Parvy et al., 2012*). Thus, oenocyte regulated lipid signalling under normal and stressful developmental conditions would be an interesting area of future research.

We functionally validated the role of the fatty acid synthase FAS1899 in CHC biosynthesis, by stably knocking down its expression during mosquito development. Oenocyte specific knock down of FAS1899 was lethal at the L2/L3 larvae stages, showing its important role for the normal mosquito development, possibly by synthesising Very Long Chain Fatty Acids (VLCFA) that are utilised at the larvae stage either for waterproofing the respiratory system (*Parvy et al., 2012*) or for other metabolic purposes. Lethality was also reported for the RNAi-mediated knock down of its ortholog (CG17374) in *D. melanogaster* before adult eclosion (*Chung et al., 2014*). Silencing of the FAS1899 expression using the polyubiquitin (Ubi) promoter also resulted in high levels of mortality (70–80%), but this time at the pupae stage and during adult emergence. This milder phenotype could be explained by the fact that the Ubi promoter drives lower levels of expression in oenocytes, which is supported also by the quantitative real time PCR data (26% knock down of the FAS1899 in adult progeny of the UAS-FAS1899i x Ubi-A10 Gal4 cross compared to the 70% of knock down seen in L2/L3 progeny of the UAS-FAS1899i x Oeno-Gal4 cross).

The relative expression levels of FAS1899 affect both the quantity and composition of CHCs produced in adult oenocytes. A 25% reduction in the total amount of hydrocarbons was observed for adults surviving knock down of FAS1899 and the CHC profile showed a decrease in the total proportion of methyl branched CHCs and an increase in saturated and un-saturated straight-chain hydrocarbons. Silencing Cyp4G16 or Cyp4G17 transcript levels in *An. gambiae* oenocytes by approximately 90% resulted in high mortality in late pupae, pharate adults and during adult emergence and, in surviving adults, a 50% reduction in the total amount of CHCs was observed (*Lynd et al., 2019*). The Cyp4G16 and Cyp4G17 P450s catalyse the final decarbonylation step in the cuticular hydrocarbon synthetic pathway, while FAS1899 is thought to catalyse the first step using acetyl-CoA to generate and elongate a fatty acyl-CoA chain. Thus, perturbing both extremes of the pathway can influence the final amount of synthesised hydrocarbons.

Partial knock down of the Desat3050 transcripts in larval oenocytes was correlated with larval lethality, similar to FAS1899 knock down. High levels of mortality were also observed when using the weaker oenocyte line (but more widespread driver line). However, no qualitative or quantitative differences in the CHC profile were observed in surviving adults. Further work is required, but it may indicate that Desat3050 catalyses the formation of unsaturated lipids that are not converted to hydrocarbons but are important in development, such that even a slight perturbation in the expression levels of this gene can have severe developmental effect. In *D. melanogaster,* either deletion or strong over-expression of the desaturase, *desat1*, result in larval mortality, showing that correct regulation of this gene is critical for development (*Köhler et al., 2009*). In addition desat1 was shown to affect not only the biosynthesis of unsaturated lipids, but also the availability of saturated lipids, as a reduction in its activity results in decreased amounts of both unsaturated and saturated fatty acids (*Ueyama et al., 2005*). Thus, a perturbation in the function of a desaturase enzyme can have a broader effect on lipid metabolism potentially leading to developmental abnormalities or lethality.

Variations in the relative abundance of CHCs on the cuticular surface have been correlated in *Anopheles* mosquitoes with species, karyotype, age and mating status (*Caputo et al., 2005*; *Polerstock et al., 2002*). Sex specific differences in the relative abundance of some CHC compounds have also been reported in *An. gambiae* (*Caputo et al., 2005*), but in contrast to other insects like *Drosophila melanogaster* (*Coyne and Oyama, 1995*), sexual dimorphism in CHCs in mosquitoes has not been reported. This lack of sex specificity is reflected in the absence of sex specific expression of CHC synthesising genes in our analysis. However, interestingly we did identify some splice variants of Cyp4G16, encoding for a different C-terminus, to be differentially expressed between male and female oenocytes, but further work is needed to validate this observation. A change in C- terminus is likely to alter the intracellular location of proteins through removal of the ER retention signal. Previous work on females has demonstrated enriched localisation of CYP4G16 on the oenocyte plasma membrane surface (*Balabanidou et al., 2016*). It would be interesting to examine males in comparison.

Variation in the abundance of CHCs has been associated in *An. coluzzii* with insecticide resistance; a 30% increase in CHC content has been correlated with a decrease in the penetration rate of pyrethroid insecticides (*Balabanidou et al., 2016*). Several of the genes implicated in CHC biosynthesis from the results of the current study are expressed at elevated levels in pyrethroid resistant mosquitoes and may provide useful genetic markers for detecting this emerging resistance phenotype. For example FAS1899 is a member of the Cyp4G16 correlation network and is over-expressed in pyrethroid insecticide resistant *An. gambiae* and *An. coluzzii* populations from Burkina Faso and Côte d'Ivoire (data from the IR-TEx web-based application [*Ingham et al., 2018*]). Thus, this gene could be implicated in cuticular resistance, through the production of a thicker cuticle with more hydrocarbons.

In addition to insecticide exposure, environmental factors can also select for changes in the CHC profile; relative proportions of unsaturated and methyl-branched CHCs altered following exposure to arid conditions in the insectary (*Reidenbach et al., 2014*) and these arid conditions were also associated with an enrichment of genes involved in lipid biosynthesis, including six elongases (*Cheng et al., 2018*), four of which overlap with the oenocyte enriched elongases identified in this study. The pleiotropic effect of alterations in CHC composition has important implications. Selection pressures that alter the CHC composition, for example the extensive use of insecticides, or an increase in aridity due to climate change, could have multiple effects on mosquito fitness and impacts on disease transmission. Investigating how the different traits influence one another and

how this is regulated by the CHC composition is a key next step to understand how mosquitoes adapt and survive in a changing environment and in response to disease control interventions.

# Materials and methods

## Key resources table

| Reagent type (species) or resource | Designation | Source or reference | Identifiers | Additional information |
|---|---|---|---|---|
| Gene (*Anopheles gambiae*) | AGAP001899 | NA | NCBI Gene ID: 1281226 | |
| Gene (*Anopheles gambiae*) | AGAP003050 | NA | NCBI Gene ID: 1272902 | |
| Genetic reagent (*An. gambiae*) | UAS-mCD8: mCherry responder line | DOI: https://doi.org/10.1016/j.ibmb.2018.03.005 | NA | Transgenic line previously generated in *Adolfi et al., 2018* and referred to as UAS-mCD8:mCherry line. Contains the coding sequence for the m-cherry fluorescent protein under the UAS promoter |
| Genetic reagent (*An. gambiae*) | Ubi-A10 Gal4 driver line | DOI: https://doi.org/10.1016/j.ibmb.2018.03.005 | NA | Transgenic line previously generated in *Adolfi et al., 2018* and referred to as PUBc-GAL4. Contains the coding sequence for the Gal4 transcription factor under the control of the *An. gambiae Polyubiquitin-c* (PUBc) gene |
| Genetic reagent (*An. gambiae*) | Oeno-Gal4 driver line | DOI: https://doi.org/10.1101/742619 | NA | Transgenic line previously generated in *Lynd et al., 2019* and referred to as A14Gal4. Contains the coding sequence for the Gal4 transcription factor under the control of an oenocyte specific enhancer |
| Genetic reagent (*An. gambiae*) | A11 docking line | DOI: https://doi.org/10.1101/742619 | NA | Transgenic line previously generated in *Lynd et al., 2019*. It is carrying two AttP docking sites flanking 3xP3:eCFP marker gene. Used for phiC31 driven RMCE of UAS-inverted repeat contructs tagged with alternative fluorescent marker gene |
| Genetic reagent (*An. gambiae*) | UAS-FAS1899i Responder line | This paper | NA | Transgenic line generated in this study carrying inverted repeats targeting the FAS1899 gene under the control of the UAS promoter |
| Genetic reagent (*An. gambiae*) | UAS-Desat3050i Responder line | This paper | NA | Transgenic line generated in this study carrying inverted repeats targeting the FAS1899 gene under the control of the UAS promoter (See Materials and methods section) |
| Recombinant DNA reagent | pSL*attB:YFP:Gyp:UAS14i:Gyp:attB (plasmid) | DOI: https://doi.org/10.1101/742619 | | Used to synthesise the inverted repeats for the generation of FAS1899i and Desat3050i transgenic lines (See Materials and methods section) |

*Continued on next page*

*Continued*

| Reagent type (species) or resource | Designation | Source or reference | Identifiers | Additional information |
|---|---|---|---|---|
| Commercial assay or kit | Arcturus PicoPure RNA Isolation Kit | Thermo Fisher | Cat. #: KIT0204 | |

## Mosquito rearing and preparation of samples for FACS

*An. gambiae* mosquitoes were reared at 28°C under 80% humidity and at a 12/12 h day/night cycle. Larvae were fed with fish food (TetraMin, Tetra GmbH), and adult mosquitoes were fed ad libitum with 10% sugar. To generate mosquitoes with fluorescent oenocytes we crossed males from the UAS-mCD8: mCherry responder line (*Adolfi et al., 2018*) with virgin females of the oeno-Gal4 driver line (*Lynd et al., 2019*). Adult progeny (2–4 days old) were collected, anesthetised on ice and dissected in 1X PBS. The head, thorax and internal tissues (midgut, malpigian tubules and reproductive tissues) were removed and the remaining integument (carcass) was cut open. Each sample (N = 12 in total, *Supplementary file 5*) consisted of 30 carcasses. Samples were washed twice with 1X PBS and incubated for 30 min at 37°C with 0,25% trypsin in 1X PBS. After incubation tissues were washed twice with 1X PBS and homogenised by pipetting up and down in 1X PBS containing 1% fetal bovine serum. Dissociated cells were filtered through a plastic filter mesh (ThermoFisher 70 μm Nylon Mesh). For samples used to isolate oenocytes (N = 6), cells were immediately used for FACS sorting. In the case of total carcass cells (N = 6) total RNA was extracted after filtering using the Arcturus PicoPure RNA extraction kit.

## FACS and RNA sequencing

For oenocyte isolation the BD ARIA III Cell Sorter (BD Biosciences) equipped with lasers at 405 and 561 nm was used. Cells were gated based on the m-Cherry fluorescence. A sample of cells from wild type G3 mosquitoes with no fluorescence was used as control to define the threshold of fluorescence for isolation. All samples were acquired in Facsdiva software version 8.1 (BD Biosciences). All debris doublets were removed from the analysis. The purity of isolation was initially assessed by visualisation of isolated cells using a Zeiss LSM 880 confocal microscope. Oenocytes were directly sorted in the extraction buffer of the Arcturus PicoPure RNA extraction kit. Total RNA was extracted based on the manufacturer's instructions, including treatment with DNAse. Generation and amplification (11 cycles) of c-DNA from all samples was done in the Center for Genome Research (University of Liverpool) using the SMART-Seq v4 Ultra Low Input RNA Kit, according to manufacturer's instructions. The cDNA samples were purified using AMPure XP beads (Beckman Coulter) and their concentration and quality determined using the Agilent 2100 Bioanalyzer and Agilent's High Sensitivity DNA Kit. Libraries were constructed with a total of 1 ng of Smarter amplified material and amplified using 12 cycles of PCR. Quality control was performed by running 1 μl undiluted library on an Agilent Technology 2100 Bioanalyzer (RRID:SCR_018043) using a High Sensitivity DNA kit. Samples were run on a Illumina HiSeq 4000 (RRID:SCR_016386).

## Pre-processing of transcriptome data

Illumina adapter sequences were removed from the read files (24 *fastq* files in total: 12 RNA-seq runs with right and left reads) using *cutadapt* 1.2.1 (*Martin, 2011*) (flag -O 3). Low-quality reads were removed using *Sickle* 1.200 (minimum window quality score of Phread = 20, removing reads shorter than 20 bp)(*Joshi and Fass, 2011*), retaining only read pairs in which both left and right reads passed quality filters. These steps were performed by the Liverpool University CGR sequencing facility. Each read file was analysed with *fastqc* 0.11.5 (*Andrews, 2014*) to confirm the absence of adapters sequences. Overall, 97.12% of reads passed the quality control process (*Supplementary file 5*).

## Genome data download

The reference gene annotation and assembly of *An. gambiae* was obtained from VectorBase (*Giraldo-Calderón et al., 2015*) (GFF and FASTA formats, version AgamP4.9).

## Gene functional annotations

We obtained the predicted peptides of each gene using *gffread* (*Geo, 2019*). Then, we annotated their Gene Ontology functional annotations using *eggNOG emapper* 1.0.3 (*Huerta-Cepas et al., 2017*) (HMM mode, which uses *hmmscan* from *HMMER* 3.2.1 (HMMER 2015)) with the euNOG database of eukaryotic protein annotations (*Huerta-Cepas et al., 2016*) (eggNOG version 4.5) as a reference. In parallel, we annotated the protein domains using Pfamscan, based on version 31 of the Pfam database (*Punta et al., 2012*).

## Analysis of differential expression

We quantified gene expression using the trimmed, clean reads. Specifically, we used *Salmon* 0.10.2 (*Patro et al., 2017*) to build an index of transcripts (*salmon index*; using the longest isoform per gene as a reference), using the quasi-mapping procedure (*–type quasi* flag) and k-mers of length 31 (*-k 31*); and then quantified transcript abundance (*salmon quant*) in each sample using the paired-end read files (using automated library type inference, *-l A* flag), in order to obtain TPM (transcripts per million) values for each gene.

Then, we performed a differential expression analysis between sample groups (female oenocytes vs female carcass cells, male oenocytes vs male carcass cells and female oenocytes vs male oenocytes) using the R *DESeq2* library 1.24.0 (*Love et al., 2014*). First, we imported the transcript quantification values from *Salmon* (see above) using the *tximport* library 1.12.0 (*Soneson et al., 2015*). Then, we performed targeted differential expression analyses between groups of samples using the *DESeq* function from *DESeq2* (using the Wald procedure for significance testing), produced a table of normalised gene counts per sample using the *counts* function (using *DESeq2* normalisation factors), and obtained the fold changes and *p*-values from a Wald test for each gene, using the *results* command (using a Benjamini-Hochberg [FDR] *p*-value correction (*Benjamini and Hochberg, 1995*) and an alpha threshold = 0.001, and all combinations of samples from Supplementary Table 1 to define the *contrast* parameter). The log-fold change values were corrected (shrunken) with *lfcShrink* and the *apeglm* algorithm (*Zhu et al., 2019*). We defined a gene as being differentially expressed in a given comparison if the adjusted $p<0.001$ and the absolute shrunken log-fold change >1 (i.e. absolute fold change >2).

We explored the variation in gene expression across samples using the normalised gene counts (log-transformed, and standardised to mean = 0 and standard deviation = 1 using the *scale* R function). First, we performed a Principal Components analysis (PCA) using the normalised gene counts of each sample (*prcomp* function of the R *stats* library).

## Heatmaps of gene expression for selected genes

To visualise changes in expression for genes involved in CHC biosynthesis, we produced heatmaps of gene expression by plotting the normalised gene counts of each gene in each sample (*pheatmap* function from the *pheatmap* 1.012 R library (Kolde 2019), using Pearson correlation values to set the order of genes).

## Analysis of alternative splicing

We used *SUPPPA2* (*Trincado et al., 2018*) to generate a set of alternative splicing events from the annotated isoforms in the *An. gambiae* genome (GFF file from Vectorbase, AgamP4.9), using the *generateEvents* mode to detect retained introns, skipped exons, and alternative first or last exons, and mutually exclusive exons (*-e SE MX RI SS FL*), with 10 bp as the minimum exon length (*-l 10*). We also calculated the expression levels at the isoform level using *Salmon* 0.10.2 (*Patro et al., 2017*) (output in TPM). Then, we used *SUPPA2 psiPerIsoform* mode to calculate the inclusion rates of each isoform (PSI: percentage spliced-in) in each sample, using the expression levels of each isoform (obtained from *Salmon*) as a reference. Differential splicing was quantified by calculating the calculating the average difference in PSI values between each sample group (male/female oenocytes and carcasses), and *p*-values were obtained using the empirical significance calculation method described in *SUPPA2* (*Trincado et al., 2018*).

The PSI values of selected differentially spliced genes ($p<0.05$) belonging to the biosynthesis pathway were reported using a heatmap table (*pheatmap* function from the *pheatmap* 1.012 R library).

## Gene functional enrichment analysis

Gene Ontology enrichments based on the GOs annotated with *eggNOG* mapper (see above) were computed using the *topGO* R library (2.34) (*Alexa and Rahnenfuhrer, 2020*). Specifically, we computed the functional enrichments based on the counts of genes belonging to the group of interest relative to all annotated genes, using Fisher's exact test and the *elim* algorithm for GO graph weighting (*Alexa et al., 2006*).

Functional enrichment tests of Pfam domain annotations were performed using hypergeometric tests as implemented in the R stats 3.6 library (*phyper*) (*R Development Core Team, 2017*), comparing the frequencies of presence of Pfam domains in a list of genes of interest to the same frequencies in the whole gene set (using unique domains per gene). We adjusted *p* values using the Benjamini-Hochberg procedure.

## Construction of phylogenetic trees

We retrieved genes belonging to gene family-members of the fatty acid biosynthesis pathway from the proteomes of *An. gambiae* (Vectorbase, AgamP4.9 annotation), *Ae. aegypti* (Vectorbase LVP_AGWG AaegL5.1 annotation) and *D. melanogaster* (Flybase r6.21 annotation). Specifically, we defined the list of candidate genes for phylogenetic analysis according to the presence of the following catalytic Pfam domains: FA_desaturase (PF00487) for desaturases (totalling 29 individual domains), ELO (PF01151) for elongases (62), NAD_binding_4 (PF07993) for reductases (61), ketoacyl-synt (PF00109) for synthases (16). Pfam annotations were obtained from Pfamscan as described above. Functional domain sequence sets were aligned using *MAFFT* 7.310 (1,000 rounds of iterative refinement, L-INS-i algorithm) (*Katoh and Standley, 2013*), and later trimmed position-wise using trimAL 1.4 (*automated1* procedure) (*Capella-Gutiérrez et al., 2009*). The trimmed alignments were used to build maximum-likelihood phylogenetic trees for each gene family, using *IQ-TREE* 1.6.10 (*Nguyen et al., 2015*). The best-fitting evolutionary model (LG substitution matrix [*Le and Gascuel, 2008*] with four Γ categories and accounting for invariant sites, or LG+I+G4) was selected for each gene family according to the BIC criterion. Phylogenetic statistical supports were calculated using the UF bootstrap procedure (1000 replicates) (*Hoang et al., 2018*).

The resulting phylogenetic trees were mid-point rooted using the *R* phangorn 2.53 library (*Schliep, 2011*), and visualisations were produced using the *phytools* 0.6–60 (*Revell, 2012*) and *ape* 5.3 libraries (plot.phylo)(*Paradis and Schliep, 2019*).

## Plasmid construction and generation of the UAS FAS1899RNAi and UAS Desat3050RNAi responder lines by PhiC31-Mediated cassette exchange

A UAS responder plasmid was generated for the expression of dsRNA targeting the third exon of the AGAP001899 gene and the first exon of the AGAP003050 gene. Specifically 200 bp inverted repeats separated by the 203 bp fourth intron of the *Drosophila melanogaster* white eye gene (CG2759) were synthesised by GeneScript and cloned into the YFP-marked responder plasmid pSL*attB:YFP:Gyp:UAS14i:Gyp:attB (*Lynd et al., 2019*) downstream of the UAS using EcoRI/NheI restriction enzymes. The intron of the *Drosophila* white eye gene was used because all internal introns of the AGAP001899 gene were shorter than 100 bp, as well as the first intron of AGAP003050, making the synthesis of the 200 bp inverted repeats impossible. Embryo injections were performed using the A11 docking line (*Lynd et al., 2019*), which carries two inverted attP sites and is marked with 3xP3-driven CFP. 350 ng/μL of the responder plasmid and 150 ng/μL of the integrase helper plasmid pKC40 encoding the phiC31 integrase (*Ringrose, 2009*) were injected as described in *Pondeville et al., 2014*. Emerging F0 individuals were outcrossed with wild type G3 individuals of the opposite sex. The F1 generation was screened for the expression of the YFP marker in the eyes and nerve cord and the absence of the CFP marker, indicating the successful cassette exchange. The direction of the cassette exchange was determined as described in *Adolfi et al., 2019* and shown to be of the A orientation. The FAS1899 RNAi and Desat3050 RNAi responder lines that were established were kept as a mix of homozygous and heterozygous individuals so as to obtain Gal4/+ progeny after crossing with the Gal4 driver lines and obtain siblings that serve as transgenic blank controls.

## Crosses of transgenic lines and qRT PCR for gene expression analysis

Crosses were performed between the responder lines UAS-FAS1899i, UAS-Desat3050i and the two Gal4 lines: oenocyte specific-GAL4 (Oeno-Gal4) (*Lynd et al., 2019*) and Ubi-A10 Gal4 line (*Adolfi et al., 2018*). Progeny (at least 10 individuals for each group, pooled in 2–3 biological replicates) of these crosses, YFP marked and blank (control siblings), were collected either at the L2-L3 stage (for the cross with the oeno-Gal4 line) or at the adult stage (for the cross with the Ubi-A10 line) and used to extract RNA with the PicoPure RNA isolation kit (Thermo Fisher Scientific) and treated with DNase using the Qiagen RNase-free DNase kit. 2ugr of RNA were reverse transcribed using SuperScript III (Invitrogen) and oligo(dT)20 primers to produce cDNA. Expression of AGAP001899 (FAS1899) and AGAP003050 (Desat3050) was validated by qPCR using the following primers: (FAS1899 Forward: 5′-AGCGATCTGCGTGATGTACC-3′, FAS1899 Reverse: 5′-GCCTTCC TCCTTAAACCCGTC-3′, Desat3050 Forward: 5′ CCGTACTACAGCGACAAGGAC-3′, Desat3050 Reverse 5′- GAACATCACAATACCGTCCGC-3′) and reference gene for normalisation the Ribosomal S7 (AGAP010592) (Forward: 5′-AGAACCAGCAGACCACCATC-3′ Reverse: 5′-GCTGCAAAC TTCGGCTATTC-3′). Expression analysis was performed according to *Pfaffl, 2001*.

## Extraction of cuticular hydrocarbons and analysis with GC-MS

CHCs were extracted from pools of adult (3–5 days old) mosquitoes (each pool consisted of 2–5 mosquitoes depending on availability, at least three pools per condition) by immersing them and gently agitating them, for 10 min at room temperature, in 200 µl of hexane (Sigma-Aldrich) spiked with 1 ng/ml of octadecane (Sigma-Aldrich) as internal standard. Hexane extracts were concentrated under a $N_2$ stream and 2 µl injected in a Waters GCT gas chromatograph-mass spectrometer. The GC column was a 30 m long, 0.25 mm internal diameter, 0.25 µm film thickness BPX5 (SGE). The oven temperature gradient was 50℃ to 370℃ at 10 ℃/minute and the carrier gas was helium (BOC) at a flow rate of 1 ml/minute. The scan range was *m/z* 40 to 450 Da in scan time 0.9 s. Compounds were identified based on their mass spectra in comparison to those of an alkane standard mixture (C10-C40, Merck 68281–2 ML-F), by comparison of their retention times and fragmentation patterns to published *Anopheles gambiae* CHC mass spectra (*Balabanidou et al., 2016*) and searches of the NIST mass spectrum library supplied with Waters MassLynx software (RRID:SCR_014271). Peak areas were measured manually using the peak integration tool in the Waters MassLynx software. The total amount of hydrocarbon present was calculated by summing all the peak areas measured relative to the area of the internal standard. Student's t-test was performed for the statistical analysis of differences in total CHC amount and relative abundance of CHC categories.

## Availability of data and materials

Transcriptome sequencing has been deposited in the European Nucleotide Archive (ENA), under PRJEB37240 project. All transgenic lines produced in this study will be provided by L.G upon request.

All data and code (in R) required to perform the differential expression, alternative splicing and phylogenetic analyses in this paper is available in the following Github repository: https://github.com/xgrau/oenocytes-agam (*Grau-Bové, 2020*; copy archived at https://github.com/elifesciences-publications/oenocytes-agam).

## Acknowledgements

We would like to thank Jesus Reiner (LSTM) for performing the FACS; Mark Prescott and Rob Beynon (University of Liverpool) who performed the CHC analysis at the Centre for Proteome Research, University of Liverpool; Simon Wagstaff (LSTM) for his advice on the RNAseq; Andriana Adolfi (LSTM) for providing the Ubi-A10 line; Rachel Davies (LSTM) for her assistance with mosquito rearing, Amalia Anthousi and Fraser Colman (LSTM) for long term maintenance and provision of stock driver and docking lines and Manuela Bernardi for preparation of figures. Flow cytometric sorting was performed on a BD FacsAria III funded by a Wellcome Trust Multi-User Equipment Grant (104936/Z/14/Z). This study was funded by the Wellcome Trust (Sir Henry Wellcome Postdoctoral fellowship (Grant reference number: 215894/Z/19/Z); a Director Catalyst Fund (pump priming award from LSTM) to LG.

## Additional information

### Funding

| Funder | Grant reference number | Author |
|--------|------------------------|--------|
| Wellcome Trust | Sir Henry Wellcome Postdoctoral Fellowship (215894/Z/19/Z) | Linda Grigoraki |
| Liverpool School of Tropical Medicine | Director's Catalyst Fund | Linda Grigoraki |

The funders had no role in study design, data collection and interpretation, or the decision to submit the work for publication.

### Author contributions

Linda Grigoraki, Conceptualization, Formal analysis, Funding acquisition, Investigation, Writing - original draft, Writing - review and editing; Xavier Grau-Bové, Resources, Formal analysis, Writing - original draft, Writing - review and editing; Henrietta Carrington Yates, Investigation, Substantial contribution to acquisition of data; Gareth J Lycett, Conceptualization, Resources, Methodology, Writing - review and editing; Hilary Ranson, Conceptualization, Resources, Writing - original draft, Writing - review and editing

### Author ORCIDs

Linda Grigoraki  https://orcid.org/0000-0001-8997-0406
Xavier Grau-Bové  https://orcid.org/0000-0003-1978-5824
Henrietta Carrington Yates  http://orcid.org/0000-0001-6199-7009
Gareth J Lycett  https://orcid.org/0000-0002-2422-053X
Hilary Ranson  https://orcid.org/0000-0003-2332-8247

### Decision letter and Author response

Decision letter https://doi.org/10.7554/eLife.58019.sa1
Author response https://doi.org/10.7554/eLife.58019.sa2

## Additional files

### Supplementary files

• Supplementary file 1. Genes differentially expressed in female oenocytes vs female total carcass cells (sheet 1); in male oenocytes vs male total carcass cells (sheet 2) and genes commonly over-expressed in female and male oenocytes compared to female and male total carcass cells (sheet 3).

• Supplementary file 2. Genes, members of gene families implicated in CHC biosynthesis and over-expressed in oenocytes, ranked in order of highest to lowest expression in Female Oenocytes. Their differential expression ($Log_2$Fold change) compared to female carcass cells is also shown. Genes above the double line are within the 200 most highly expressed genes.

• Supplementary file 3. Genes, members of gene families implicated in CHC biosynthesis and over-expressed in oenocytes, ranked in order of highest to lowest expression in Male Oenocytes. Their differential expression ($Log_2$Fold change) compared to male carcass cells is also shown. Genes above the double line are within the 200 most highly expressed genes.

• Supplementary file 4. Genes differentially expressed in female oenocytes vs male oenocytes.

• Supplementary file 5. Samples used for Illumina RNAseq. Number of raw reads produced for each sample and number of reads after quality control.

• Transparent reporting form

## Data availability

Transcriptome sequencing has been deposited in the European Nucleotide Archive (ENA), under PRJEB37240 project. All data generated or analysed during this study are included in the manuscript and supporting files. Source data files have been provided for Figures 1, 2, 3 and 5.

The following dataset was generated:

| Author(s) | Year | Dataset title | Dataset URL | Database and Identifier |
|---|---|---|---|---|
| Grau-Bové X, Grigoraki L | 2020 | Transcriptome of Anopheles gambiae oenocytes | https://www.ebi.ac.uk/ena/data/search?query=PRJEB37240 | European Nucleotide Archive, PRJEB37240 |

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

Appendix 1

## Differential expression of splice isoforms in oenocytes

We investigated whether specific gene isoforms are differentially expressed in oenocytes (at a p<0.05). 672 genes were found to have at least one isoform differentially expressed in female oenocytes compared to female carcass cells and 752 to have at least one isoform differentially expressed in male oenocytes compare to male carcass cells. The same analysis was performed for female and male oenocytes showing 578 genes to have at least one isoform differentially expressed between sexes (*Appendix 1—figure 1A*). Five genes belonging to the six gene families implicated in the hydrocarbon biosynthetic pathway (the elongase AGAP004373, the desaturases AGAP003051, AGAP004572 and AGAP01713 and the decarbonylase P450 Cyp4G16) had at least one isoform differentially expressed in at least two of the three comparisons (female oenocytes vs female total carcass cells, male oenocytes vs male total carcass cells and female vs male oenocytes) (*Appendix 1—figure 1B*). Most of the isoforms for these genes differ solely in the untranslated regions. Exceptions are the RD isoform of Cyp4G16 that encodes for a slightly truncated protein with a different C-terminus (last 19 a.a) compared to the other isoforms, and isoforms RA and RB of the desaturase AGAP003051, which encode proteins with highly diverged C-termini. We need however to point out that the two predicted isoforms for AGAP003051 might be affected by some annotation error, as the AGAP003051-RB isoform is identical with the adjacent AGAP003050 transcript after nucleotide 407 (total length of 1038 nt). The AGAP003051-RB isoform was more abundant in oenocytes compared to total Carcass cells and more abundant in female oenocytes compared to male oenocytes, although this latter difference was clearly driven by one of the male oenocyte replicates. The Cyp4G16-RD isoform was enriched in female oenocytes in comparison to both female carcass cells and male oenocytes (*Appendix 1—figure 1B*).

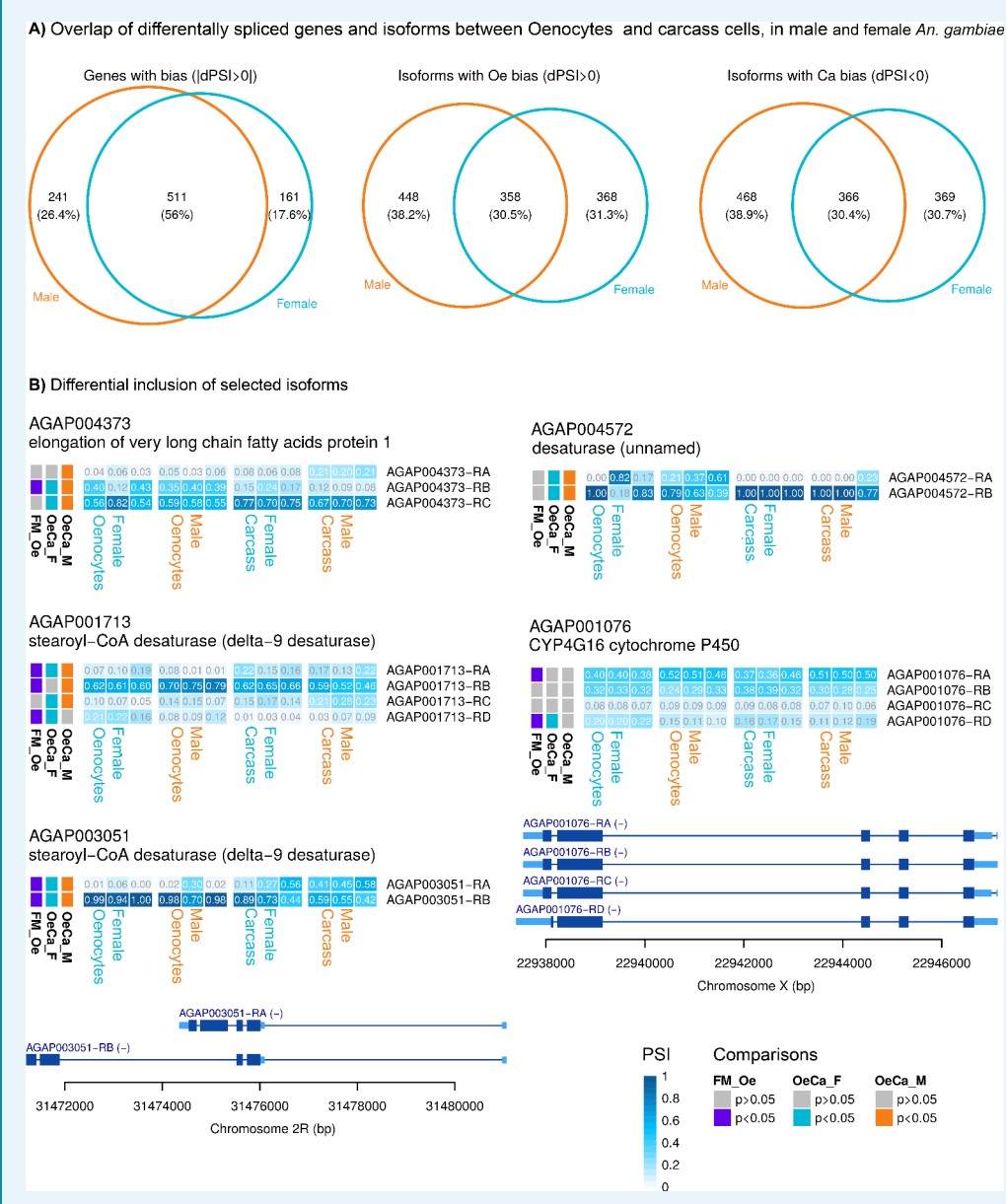

**Appendix 1—figure 1.** Splice variant analysis. (**A**) Venn diagram (on the left) showing the number of genes with differential expression of at least one isoform in female and/or male oenocytes compared to total carcass cells. Venn diagrams showing the number of gene isoforms with enriched (middle diagram) or reduced (right diagram) expression in female and/or male oenocytes compared to total carcass cells. (**B**) Heat maps showing the frequency (PSI) of isoforms (in each sample used for RNAseq) for genes belonging to gene families implicated in CHC biosynthesis. Isoforms that encode for different proteins are depicted. Comparisons performed: Female Oenocytes vs Female total carcass cells (OeCa_F), Male Oenocytes vs Male total carcass cells (OeCa_M), Female Oenocytes vs Male Oenocytes (FM_Oe). Source data: *Supplementary file 2*.

The online version of this article includes the following source data is available for figure 1:

**Appendix 1—figure 1—source data 1.** Genes showing isoform specific differential expression in oenocytes.

