## [Decision Letter]

**Acceptance summary:**

This study provides new insights into the enzyme machinery involved in the biosynthesis of outer layer cuticle hydrocarbons in the *Anopheles* mosquito. Candidate hydrocarbon biosynthetic genes were identified following transcriptional analysis of oenocytes, specialized secretory cells in the subdermis, and the function of a putative fatty acid synthase validated by genetic knock-down studies. This approach can be used to identify other genes involved in cuticle biogenesis with important implications for understanding the biology and the malarial vectorial capacity of *Anopheles* mosquitoes.

**Decision letter after peer review:**

Thank you for submitting your article "Cuticular hydrocarbon biosynthesis in malaria vectors: insights from the adult oenocyte transcriptome" for consideration by *eLife*. Your article has been reviewed by three peer reviewers, including Malcolm J McConville as the Reviewing Editor and Reviewer #1, and the evaluation has been overseen by Michael Marletta as the Senior Editor. The following individual involved in review of your submission has agreed to reveal their identity: Nora Besansky (Reviewer #3).

The reviewers have discussed the reviews with one another and the Reviewing Editor has drafted this decision to help you prepare a revised submission.

Summary:

This study provides new insights into the enzyme machinery involved in the biosynthesis of outer layer cuticle hydrocarbons in the *Anopheles* mosquito. Cuticle hydrocarbons are synthesized in oenocytes, specialized secretory cells in the subdermis. These cells were purified from a transgenic *Anopheles gambiae* line containing fluorescently tagged oenocytes and RNAseq used to identify candidate genes for each step in cuticle hydrocarbon biosynthesis. Genetic knock-down of an oenocyte-enriched fatty acid synthase supported the role of this enzyme in cuticle hydrocarbon biosynthesis. The study represents a rich resource for further studies delineating the role of variable hydrocarbon composition on the biology and the malarial vectorial capacity of *Anopheles* mosquitoes.

Essential revisions:

1) Please better explain possible reasons for the Ubi-gal4 x uas.desat30501 no significant silencing/no change in CHC, but mortality, related to the oenocytes.

2) Please provide better images if available showing adulty mortality in the supplement (not clear if they suffer from desiccation).

3) Is there any possible explanation, why 58% of the RNAseq transcripts map top the genome? (compared to other studies)

4) Could the authors possibly expand on the possibility that oenocytes have roles in neurogenesis during *Drosophila* embryonic development.

---

## [Author Response]

Essential revisions:1) Please better explain possible reasons for the Ubi-gal4 x uas.desat30501 no significant silencing/no change in CHC, but mortality, related to the oenocytes.

We added additional text in the Discussion (eighth paragraph) expanding on possible reasons for the mortality seen in progeny of the Ubi-Gal4 x UAS Desat3050 cross, despite the absence of clear changes in their cuticular hydrocarbon profile. Whilst we cannot be certain for the reasons behind this observation, we highlight examples from *Drosophila* where perturbation of desaturase levels affected development.

2) Please provide better images if available showing adulty mortality in the supplement (not clear if they suffer from desiccation).

Whilst, under normal circumstances, we would be very happy to comply with this request, this is extremely challenging under current working conditions. Unfortunately we don’t have alternative images showing the adult mortality of the Desat3050i and FAS1899i individuals available. Hence, to obtain these we would have to set up new crosses and rear their progeny to adulthood, which would take at least four weeks of extra work, something quite difficult at the moment due to restrictions imposed to our work by the Covid-19 pandemic. We would kindly request to retain the original supplementary figures. We do accept that the quality could be improved, but the originals do clearly show the high mortality of individuals following knock-down of the two studied genes, in the pupae and adult stages, compared to wild type individuals. It is most possible that mortality is related to desiccation as the reviewer suggested, although this would best be tested by monitoring water loss before a definitive statement could be made.

3) Is there any possible explanation, why 58% of the RNAseq transcripts map top the genome? (compared to other studies)

We attributed this relatively low mapping rate to a combination of factors.

1) Our read sets contained a certain amount of short reads that, whilst having high-quality base calls, could not be aligned to the predicted transcriptome. Specifically, we used a stringent mapping procedure that relied on a minimum alignment length ofk=31 base pairs so as to take advantage of our long reads (average length = 150 bp), as per the recommended good practice in thesalmonaligner. However, as part of read quality pre-processing, we shortened reads by removing low-quality base calls (Phred < 20), which resulted in a small amount of reads pairs with one pair shorter thank=31 base pairs (~2% reads were <31 bp) A similar effect might originate from reads that retain the transcript's poly-A tails (a further ~1% reads per sample). These two groups of reads are not derived from sequencing errors, but they cannot be aligned and thus drive down mapping rates. While they are often removed from RNA-seq libraries because they can hinder transcriptome assemblies, this is not a problem in our analyses because we relied on the high-quality gene annotations from the*Anopheles gambiae*genome.

2) It is also worth mentioning thatthesalmonaligner relies on mapping reads to a set of spliced, predicted transcripts instead of the genome. Thus, any reads containing retained introns due to alternative splicing would not align. This is an important difference with commonly used read mappers that support spliced reads on the genome, such asSTAR.

3) Finally, another possible cause of lowered mapping rates is the high frequency of single nucleotide polymorphisms in*Anopheles gambiae*genomes, where up to 30% of genomic bases have variants (Miles et al. 2017, Nature doi:10.1038/nature24995). To test the effect of genetic distance to the reference on mapping rates, we have aligned our reads to the contigs from ade novoassembled transcriptome that could be aligned to the reference transcriptome (same alignment parameters as in the paper), and we have observed ~25% higher mapping rates across all samples.

Given the increased mapping rates observed in our tests against ade novoassembled transcriptome (point 3), it is worth emphasising our reasons for using genome-derived gene models in our main analyses:

1) by doing so we are able to interpret our results using the curated gene models available for*Anopheles gambiae*rather than often-fragmented assembled transcripts;

2) in principle, low mapping rates in SNP-rich transcripts would affect all samples equally and thus not alter the results of our differential expression analyses; and

3) in fact, we found no indication that SNP-rich transcripts had lower mapping rates in our samples. Specifically, there was no significant negative correlation between mapping rates (either measured as TPMs or number of mapped reads per transcript) and SNP density (using SNPs from coding sequences retrieved from the *Anopheles gambiae* 1000 genomes project), according to a Pearson's correlation test (significance threshold p<0.01).

4) Could the authors possibly expand on the possibility that oenocytes have roles in neurogenesis during Drosophila embryonic development.

We added an extra paragraph in the Discussion (fourth paragraph) expanding on the link between oenocytes and nervous system development in *D. melanogaster*. We have also expanded on oenocyte’s potential role in lipid, synthesis and secretion (signalling), which is another function suggested by the biological processes enriched in our oenocyte transcriptome dataset.